# A simple model of recognition and recall memory

**Nisheeth Srivastava**
Computer Science, IIT Kanpur
Kanpur, UP 208016
`nsrivast@cse.iitk.ac.in`

**Edward Vul**
Dept of Psychology, UCSD
9500 Gilman Drive La Jolla CA 92093
`evul@ucsd.edu`

## Abstract

We show that several striking differences in memory performance between recognition and recall tasks are explained by an ecological bias endemic in classic memory experiments - that such experiments universally involve more stimuli than retrieval cues. We show that while it is sensible to think of recall as simply retrieving items when probed with a cue - typically the item list itself - it is better to think of recognition as retrieving cues when probed with items. To test this theory, by manipulating the number of items and cues in a memory experiment, we show a crossover effect in memory performance within subjects such that recognition performance is superior to recall performance when the number of items is greater than the number of cues and recall performance is better than recognition when the converse holds. We build a simple computational model around this theory, using sampling to approximate an ideal Bayesian observer encoding and retrieving situational co-occurrence frequencies of stimuli and retrieval cues. This model robustly reproduces a number of dissociations in recognition and recall previously used to argue for dual-process accounts of declarative memory.

## 1   Introduction

Over half a century, differences in memory performance in recognition and recall-based experiments have been a prominent nexus of controversy and confusion. There is broad agreement among memory researchers, following Mandler's influential lead, that there are at least two different types of memory *activities* - recollection, wherein we simply remember something we want to remember, and familiarity, wherein we remember having seen something before, but nothing more beyond it [8]. Recall-based experiments are obvious representatives of recollection. Mandler suggested that recognition was a good example of familiarity activity.

Dual-process accounts of memory question Mandler's premise that recognition is exclusively a familiarity operation. They argue, phenomenologically, that recognition could also succeed successful recollection, making the process a dual composition of recollection and familiarity [20]. Experimental procedures and analysis methods have been designed to test for the relative presence of both processes in recognition experiments, with variable success. These endeavors contrast with strength-based single-process models of memory that treat recognition as the retrieval of a weak trace of item memory, and recall as retrieval of a stronger trace of the same item [19].

The single/dual process dispute also spills over into the computational modeling of memory. Gillund and Shiffrin's influential SAM model is a single-process account of both recognition and recall [4]. In SAM and other strength-based models of declarative memory, recognition is modeled as item-relevant associative activation of memory breaching a threshold, while recall is modeled as sampling items from memory using the relative magnitudes of these associative activations. In contrast, McClelland's equally influential CLS model is explicitly a dual-process model, where a fast learning hippocampal component primarily responsible for recollection sits atop a slow learning neocortical component responsible for familiarity [9]. Wixted's signal detection model tries to bridge the gap between

these accounts by allowing dual process contributions to combine additively into a unidimensional strength variable [19]. While such pragmatic syntheses are useful, the field is still looking for a more satisfactory theoretical unification.

The depth of the difference between the postulated dual processes of recollection and familiarity depends inevitably on the strength of the quantitative and qualitative dissociations that previous research has documented in memory tasks, prominent among which are recognition and recall. Mandler, for instance, postulated a one-to-one mapping between recognition and familarity on one hand and recall and recollection on the other [8], although other authors hold more nuanced views [20]. Notwithstanding such differences of opinion, the road to discovering useful single-process accounts of declarative memory has to go through explaining the multiple performance dissociations between recognition and recall memory tasks. To the extent that single process accounts of both tasks can explain such dissociations, differences between recollection and familiarity will not seem nearly as fundamental.

Improved strength-based models have competently modeled a large array of recognition-recall dissociations [13], but fail, or have to make intricate assumptions, in the face of others [20]. More importantly, the SAM model and its descendants are not purely single-process models. They model recognition as a threshold event and recall as a sampling event, with the unification coming from the fact that both events occur using the same information base of associative activation magnitude. We present a much simpler single process model that capably reproduces many critical qualitative recognition-recall dissociations. In the process, we rationalize the erstwhile abstract associative activation of strength-based memory models as statistically efficient monitoring of environmental co-occurrence frequencies. Finally, we show using simulations and a behavioral experiment, that the large differences between recognition and recall in the literature can be explained by the responses of an approximately Bayesian observer tracking these frequencies to two different questions.

## 2 Model

We use a very simple model, specified completely by heavily stylized encoding and retrieval processes. The encoding component of our model simply learns the relative frequencies with which specific conjunctions of objects are attended to in the world. We consider objects $x$ of only two types: items $x_i$ and lists $x_l$. We model each timestep as as a Bernoulli trial between the propensity to attend to any of the set of items or to the item-list itself, with a uniform prior probability of sampling any of the objects. Observers update the probability of co-occurrence, defined in our case rigidly as 1-back occurrence, inductively as the items on the list are presented. We model this as the observer's sequential Bayesian updates of the probability $p(\mathbf{x})$, stored at every time step as a discrete memory engram $m$.

Thus, in this encoding model, information about the displayed list of items is available in distributed form in memory as $p(x_i, x_l|m)$, with each engram $m$ storing one instance of co-occurrence. The true joint distribution of observed items,to the extent that it is encoded within the set of all task-relevant memory engrams $\mathcal{M}$ is then expressible as a simple probabilistic marginalization,

$$p(x_i, x_l) = \sum_{m \in \mathcal{M}} p(x_i, x_l|m)p(m),\tag{1}$$

where we assume that $p(m)$ is flat over $\mathcal{M}$, i.e. we assume that within the set of memory engrams relevant for the retrieval cue, memory access is random.

Our retrieval model is approximately Bayesian. It assumes that people sample a small subset of all relevant engrams $\mathcal{M}' \subset \mathcal{M}$ when making memory judgments. Thus, the joint distribution accessible to the observer during retrieval becomes a function of the set of engrams actually retrieved,

$$p_{\mathcal{M}_k}(x_i, x_l) = \sum_{m \in \mathcal{M}_k} p(x_i, x_l|m)p(m),\tag{2}$$

where $\mathcal{M}_k$ denotes the set of first $k$ engrams retrieved.

Following a common approach to sampling termination in strength-based sequential sampling memory models, we use a novelty threshold that allows the memory retrieval process to self-terminate when incoming engrams no longer convey significantly novel information [4, 13]. We treat the arrival of the

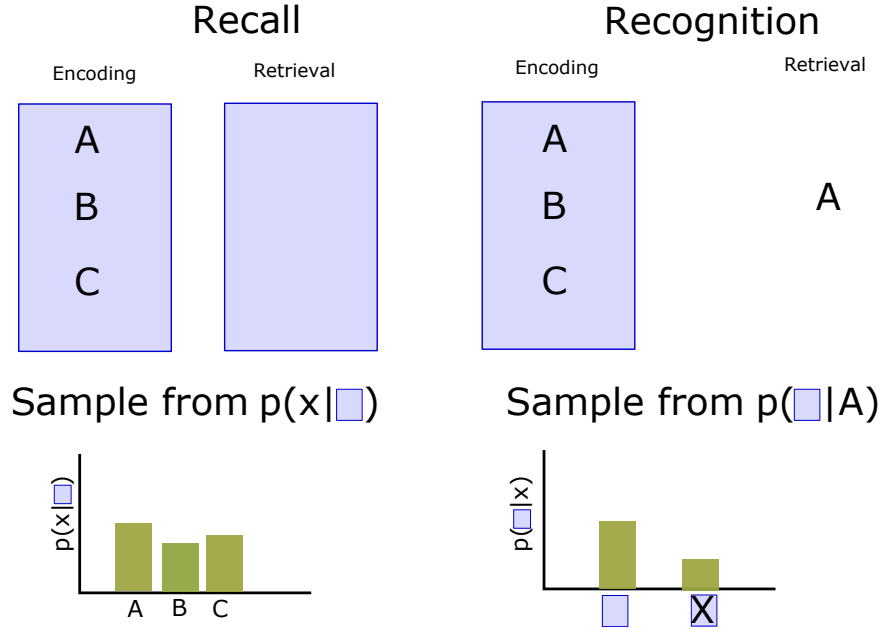

Figure 1: Illustrating the ecological difference in retrieval during recognition and recall memory experiments. We model recall retrieval as a probabilistic query about items conditioned on the item list and recognition retrieval as a probabilistic query about the item list conditioned on the item presented during retrieval. Since there are almost always more items than lists in classic memory experiments, the second conditional distribution tends to be formed on a smaller discrete support set than the former.

$k^{th}$ successive engram into working memory as a signal for the observer to probabilistically sample from $p_{\mathcal{M}_k}$. The stopping rule for memory retrieval in our model is for $n$ consecutive identical samples being drawn in succession during this internal sampling, $n$ remaining a free parameter in the model. This rule is designed to capture the fact that memory search is frugal and self-terminating [15]. The sample drawn at the instant the novelty threshold is breached is overtly recalled. Since this sample is drawn from a distribution constructed by approximately reconstructing the true encoded distribution of situational co-occurrences, the retrieval model is approximately Bayesian. Finally, since our encoding model ensures that the observer knows the joint distribution of event co-occurrences, which contains all the information needed to compute marginals and conditionals also, we further assume that these derivative distributions can also be sampled, using the same retrieval model, when required.

We show in this paper that this simple memory model yields both recognition and recall behavior. The difference between recognition and recall is simply that these two retrieval modalities ask two different questions of the same base of encoded memory - the joint distribution $p(x_i, x_l)$. We illustrate this difference in Figure 1. During recall-based retrieval, experimenters ask participants to remember all the items that were on a previously studied list. In this case, the probabilistic question being asked is 'given $x_l$, find $x_i$', which our model would answer by sampling $p(x_i|x_l)$. In item-recognition experiments, experimenters ask participants to determine whether each of several items was on a previously shown list or not. We assert that in this case the probabilistic question being asked is 'given $x_i$. find $x_l$', which our model would answer by sampling $p(x_l|x_i)$.

Our operationalization of recognition as a question about the list rather than the item runs contrary to previous formalizations, which have tended to model it as the associative activation engendered in the brain by observing a previously seen stimulus - models of recognition memory assume that the activation for previously seen stimuli is greater, for all sorts of reasons. In contrast, recall is modeled in classical memory accounts much the same way as in ours - as a conditional activation of items associated with retrieval cues, including both the item list and temporally contiguous items. Our approach assumes that the same mechanism of conditional activation occurs in recognition as well - the difference is that we condition on the item itself.

## 3  Basic prediction: fast recognition and slow recall

The sample-based threshold used to terminate memory retrieval in our model $\epsilon$ does not depend on the size of the support of the probability distribution being sampled from. This immediately implies that, for the same threshold sample value, the model will take longer to approach it when sampling from a distribution with larger support than when sampling from distributions with smaller support.

In classical memory experiments, observers are typically asked to memorize multiple items associated with one, or a few, lists. Thus, there is an ecological bias built into classic memory experiments such that $|items| \gg |lists|$. Making this assumption immediately rationalizes the apparent difference in speed and effort between recognition and recall in our model. Because the recognition task samples $p(list|item)$, its sample complexity is lower than recall, which involves sampling $p(item|list)$ from memory.

To verify this numerically, starting from identical memory encodings in both cases, we ran 1000 simulations of recognition and recall respectively using our retrieval model, using a fixed $n = 5$ [1]. The results, measured in terms of the number of retrieval samples $k$ drawn before termination in each of the 1000 trials, are shown in the left panel of Figure 2. The sample complexity of recall is evidently higher than for recognition[2]. Thus, we suggest that the fundamental difference between recognition and recall - that recognition is easier and recall is harder - is explicable simply by virtue of the ecological bias of memory experiments that use fewer cues than stimuli.

The difference in speed between recollection and familiarity processes, as measured in recall and recognition experiments, has been one of the fundamental motivations for proposing that two memory processes are involved in declarative memory. Dual-process accounts have invoked priority arguments instead, e.g. that information has to pass through semantic memory, which is responsible for recognition, before accessing episodic memory which is responsible for recall [17].Single process accounts following in the lineage of SAM [4] have explained the difference by arguing that recognition involves a single comparison of activation values to a threshold, whereas recall involves competition between multiple activations for sampling. Our model rationalizes this distinction made in SAM-style sequential sampling models by arguing that recognition memory retrieval is identical to recall memory retrieval; only the support of the distribution from which the memory trace is to be probabilistically retrieved changes. Thus, instead of using a race to threshold for recognition and a sampling process in recall, this model uses self-terminating sampling in both cases, explaining the main difference between the two tasks - easy recognition and hard recall - as a function of typical ecological parameter choices. This observation also explains the relative indifference of recognition tasks to divided attention conditions, in contrast with recall which is heavily affected [2]. Because of the lower sample complexity of recognition, fewer useful samples are needed to arrive at the correct conclusion.

## 4  An empirical test

The explanation our model offers is simple, but untested. To directly test it, we constructed a simple behavioral experiment, where we would manipulate the number of items and cues keeping the total number of presentations constant, and see how this affected memory performance in both recognition and recall retrieval modalities. Our model predicts that memory performance difficulty scales up with the size of the support of the conditional probability distribution relevant to the retrieval modality. Thus recall, which samples from $p(item|list)$, should become easier as the number of items to recall per cue reduces. Similarly recognition, which samples from $p(listitem)$, should become harder as the number of cues per item increases. Because classic memory experiments have tended to use more items than cues (lists), our model predicts that such experiments would consistently find recognition to be easier than recall. By inverting this pattern, having more cues than items, for instance, we would expect to see the opposite pattern hold. We tested for this performance crossover using the following experiment.

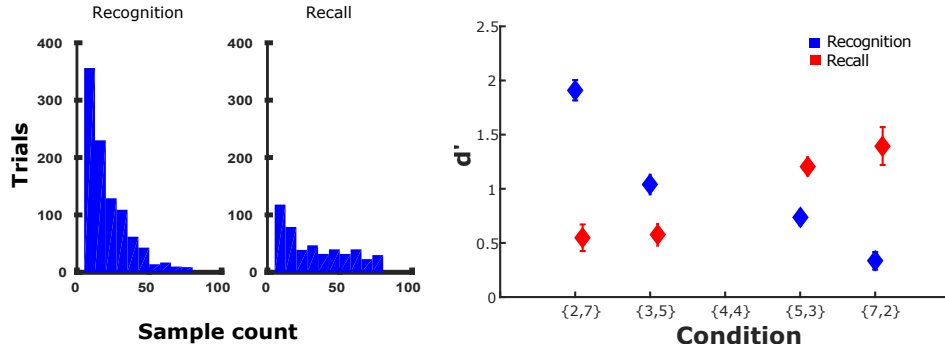

Figure 2: (Left) Simulation results show easier recognition and harder recall given typical ecological choices for stimuli and cue set sizes. (Right) Results from experiment manipulating the stimuli and cue set size ratio. By manipulating the number of stimuli and cues, we predicted that we would be able to make recall harder than recognition for experiment participants. The results support our prediction unambiguously.Error bars show s.e.m.

We used a $2 \times 2$ within subject factorial design for this experiment, testing for the effect of the retrieval mode - recognition/recall and either a stimulus heavy, or cue heavy selection of task materials. In addition, we ran two conditions between subjects, using different parameterization of the stimuli/cue ratios. In the stimulus heavy condition, for instance, participants were exposed to 5 stimuli associated with 3 cues, while for the cue heavy condition, they saw 3 stimuli associated with 5 cues. The semantic identity of the stimuli and cue sets were varied across all four conditions randomly, and the order of presentation of conditions to participants was counterbalanced. All participants worked on all four of the memory tasks, with interference avoided with the use of semantically distinct category pairs across the four conditions. Specifically, we used number-letter, vegetable-occupation, fruit-adjective and animal-place category pairs for the four conditions. Within each category, stimuli/cues for a particular presentation were sampled from a 16 item master list, such that a stimulus could not occur twice in conjunction with the same cue, but could occur in conjunction with multiple cues.

120 undergraduates participated in the experiment for course credit. Voluntary consent was obtained from all participants, and the experimental protocol was approved by an institutional IRB. We told experiment participants that they would be participating in a memory experiment, and their goal was to remember as many of the items we showed them as possible. We also told them that the experiment would have four parts, and that once they started working on a part, there would be no opportunity to take a break until it ended. 80 participants performed the experiment with 3/5 and 5/3 stimulus-to-cue mappings, 40 did it with 2/7 and 7/2 stimulus-to-cue mappings. Note that in all cases, participants saw approximately the same number of total stimulus-cue bindings (3x5 = 15 or 2x7 = 14), thus undergoing equivalent cognitive load during encoding.

Stimuli and cues were presented onscreen, with each pair appearing on the screen for 3 seconds, followed by an ITI of equal duration. To prevent mnemonic strategy use at the time of encoding, the horizontal orientation of the stimulus-cue pair was randomly selected on each trial, and participants were not told beforehand which item category would be the cue; they could only discover this at the time of retrieval[3]. Participants were permitted to begin retrieval at their own discretion once the encoding segment of the trial had concluded within each condition. All participants chose to commence retrieval without delay. Participants were also permitted to take breaks of between 2-5 minutes between working on the different conditions, with several choosing to do so.

Once participants had seen all item-pairs for one of the conditions, the experiment prompted them to, when ready, click on a button to proceed to the testing phase. In the recall condition, they saw a text box and a sentence asking them to recall all the items that occurred alongside item X, where X was randomly chosen from the set of possible cues for that condition; they responded by typing in the words they remembered. For recognition, participants saw a sentence asking them to identify if X had occurred alongside Y, where Y was randomly chosen from the set of possible cues for that condition.

After each forced yes/no response, a new X was shown. Half the X's shown in the recognition test were 'lures', they had not been originally displayed alongside Y.

Memory performance was measured using d', which is simply the difference between the z-normed hit rate and false alarm rate, as is conventional in recognition experiments. d' is generally not used to measure recall performance, since the number of true negatives is undefined in classic recall experiments, which leaves the false alarm rate undefined as well. In our setup, the number of true negatives is obviously the number of stimuli the participant saw that were not on the specific list being probed, which is what we used to calculate d-prime for recall as well.

The right panel in Figure 2 illustrates the results of our experiment. The predicted crossover is unambiguously observed. Further, changes in memory performance across the stimulus-cue set size manipulation is symmetric across recognition and recall. This is precisely what we'd expect if set size dependence was symmetrically affecting memory performance across both tasks as occurs in our model. While not wishing to read too much into the symmetry of the quantitative result, we note that such symmetry under a simple manipulation of the retrieval conditions appears to suggest that the manipulation does in fact affect memory performance very strongly. Overall, the data strongly supports our thesis - that quantitative differences in memory performance in recognition and recall tasks are driven by differences in the set size of the underlying memory distribution being sampled. The set size of the distribution being sampled, in turn, is determined by task constraints - and ends up being symmetric when comparing single-item recognition with cued recall.

## 5 Predicting more recognition-recall dissociations

The fact that recognition is usually easier than recall - more accurate and quicker for the same stimuli sets - is simply the most prominent difference between the two paradigms. Experimentalists have uncovered a number of interesting manipulations in memory experiments that affect performance on these tasks differentially. These are called recognition-recall dissociations, and are prominent challenges to single-process accounts of the two tasks. Why should a manipulation affect only one task and not the other if they are both outcomes of the same underlying process? [20] Previous single-process accounts have had success in explaining some such dissociations. We focus here on some that have proved relatively hard to explain without making inelegant dissociation-specific assumptions in earlier accounts [13].

### 5.1 List strength effects and part set cuing

Unidimensional strength-based models of memory like SAM and REM fail to predict the list strength effect [12] where participants' memory performance in free recall is lower than a controlled baseline for weaker items on mixed lists (lists containing both strongly and weakly encoded items). Such behavior is predicted easily by strength-based models. What they find difficult to explain is that performance does not deviate from baseline in recognition tasks. The classical explanation for this discrepancy is the use of a *differentiation* assumption. It is assumed that stronger items are associated more strongly to the encoding context, however differences between the item itself as shown, and its encoded image are also stronger. In free recall, this second interaction does not have an effect, since the item itself is not presented, so a positive list strength effect is seen. In recognition, it is conjectured that the two influences cancel each other out, resulting in a null list strength effect [13].

A lot of intricate assumptions have to hold for the differentiation account to hold. Our model has a much simpler explanation for the null list-strength effect in recognition. Recognition involves sampling based on the strength of the associative activation of the list given a *specific* item and so is independent of the encoding strength of other items. On the other hand, recall involves sampling from $p(item|list)$ across all items, in which case, having a distribution favoring other items will reduce the probability that the unstrengthened items will be sampled. Thus, the difference in which variable the retrieval operation conditions on explains the respective presence and absence of a list strength effect in recall and recognition.

The left panel in Figure 3 presents simulation results from our model reproducing this effect, where we implement mixed lists by presenting certain stimuli more frequently during encoding and retrieve in the usual manner. Hit rates are calculated for less frequently presented stimuli. For either elicitation modality, the actual outcome of the retrieval itself is sampled from the appropriate conditional

distribution as a specific item/cue. In this particular experiment, which manipulates how much training observers have on some of the items on the list, the histories entering the simulation are generated such that some items co-occur with the future retrieval cue more frequently than others, i.e. two items occur with a probability of 0.4 and 0.3 respectively, and three items occur with a probability of 0.1 each alongside the cue. The simulation shows a positive list strength effect for recall (weaker hit rates for less studied items) and a null list strength effect for recognition, congruent with data.

Our model also reconciles the results of [1] who demonstrated that the list strength effect does not occur if we examine only items that are the first in their category to be retrieved. For category-insensitive strength-based accounts, this is a serious problem. For our account, which is explicitly concerned with how observers co-encode stimuli and retrieval cues, this result is no great mystery. For multi-category memory tests, the presence of each semantic category instantiates a novel list during encoding, such that the strength-dependent updates during retrieval apply to each individual $p(item|list)$ and do not apply across the other category lists.

More generally, the dynamic nature of the sampled distribution in our Bayesian theory accommodates the theoretical views of both champions of strength-dependent activation and retrieval-dependent suppression [1]. Strength-dependent activation is present in our model in the form of the Bayesian posterior over cue-relevant targets at the time when cued recall commences; retrieval-dependent suppression of competitors is present in the form of normalization of the distribution during further sequential Bayesian updates as the retrieval process continues. Assigning credit differentially to individual categories predicts an attenuation (though not removal) of the list strength effect, due to the absence of learning-induced changes for the first-tested items, as well diminishing memory performance with testing position seen in [1].

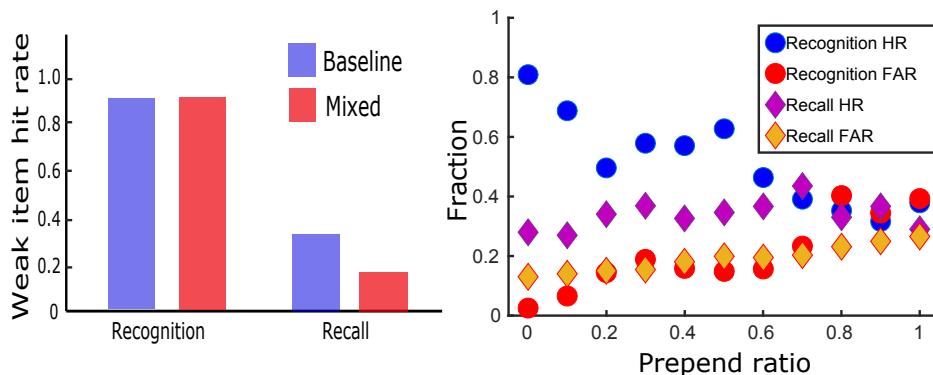

Figure 3: Reproducing (left) list strength effects and (right) the word frequency mirror effect using our model.

The part set cueing effect is the observation that showing participants a subset of the items to be recalled during retrieval reduces their recall performance for non-shown items [11]. This effect does not appear in recognition experiments, which is again problematic for unidimensional strength-based memory models. Our model has a simple explanation. The presented items during retrieval are simply treated as further encoding opportunities for the seen items, resulting in a list strength imbalance as above. This affects recall, but not recognition for the same reasons the list strength effect does.

## 5.2  Mirror effect

Another interesting effect that strength-based memory models have found hard to explain is the word-frequency mirror effect [5]. This is seen when participants see two different classes of items in recognition experiments. It is found, for instance, that unique items are both recognized more accurately as previously seen and unseen in such experiments than common items. Such a pattern of memory performance is contrary to the predictions of nearly all accounts of memory that depend on unidimensional measures of memory strength, who can only model adaptive changes in memory performance via shifts in the response criterion [19] that do not permit both the hit rate and the false alarm rate to improve simultaneously.

The essential insight of the mirror effect is that some types of stimuli are intrinsically more memorable than others, a common-sense observation that has proved surprisingly difficult for strength-based memory models to assimilate.This difficulty extends to our own model also, but our inductive framework allows us to express the assumptions about information that the stimuli base frequency adds to the picture in a clean way. Specifically, in our model observers use $p(list|item)$ for recognition, which is high for unique items and low for common items by Bayesian inversion because $p(item|list)/p(item) \approx 1$ for unique items, because they are unlikely to have been encountered outside the experimental context, and $\ll 1$ for common items. In contrast, observers sample from $p(item|list)$ during recall, removing the effect of the frequency base rate $p(item)$, so that the pattern of results is inverted: performance is equivalent or better than baseline for common stimuli than for rare ones [6], since they are more likely to be retrieved in general.

The right panel in Figure 3 shows simulation results using our model wherein we used two possible cues during encoding, one to test performance during retrieval and one to modify the non-retrieval frequency of stimuli encounters. For this experiment, which manipulates where we have to influence how often the relative frequency with which the observers have seen the items in task-irrelevant contexts other than the retrieval task, we prepended the base case history (of size 50 time steps) with differently sized prior history samples (between 10 and 50 time steps long, in steps of 5), wherein items co-occurred with cues that were not used during retrieval. The simulation results show that, in recognition, hit rates drop and false alarm rates rise with more exposure to items outside the experimental list context (high frequency items). Since our model assumes unambiguous cue conditioning, it predicts unchanged performance from baseline for recall. More intricate models that permit cue-cue associations may reproduce the advantage for common items documented empirically.

## 6 Discussion

We have made a very simple proposal in this paper. We join multiple previous authors in arguing that memory retrieval in cued recall tasks can be interpreted as a question about the likelihood of retrieving an item given the retrieval cue, typically the list of items given at the time of encoding [17, 8, 4]. We depart from previous authors in arguing that memory retrieval in item recognition tasks asks the precisely opposite question: what is the likelihood of a given item having been associated with the list? We integrated this insight into a simple inference-based model of memory encoding, which shares its formal motivations with recent inference-based models of conditioning [3, 14], and an approximately Bayesian model of memory retrieval, which samples memory frugally along lines motivated on information-theoretic [18] and ecological grounds [16] by recent work.

Our model is meant to be expository and ignores several large issues that other richer models typically engage with. For instance, it is silent about the time decay of memory particles, the partitioning of the world into items and cues, and how it would go about explaining other more intricate memory tasks like plurality discrimination and remember-know judgments. These omissions are deliberate, in the sense that we wanted to present a minimal model to deliver the core intuition behind our approach - that differences in memory performance in recognition and recall are attributable to no deeper issue than an ecological preference to test memory using more items than lists. This observation can now subsequently guide and constrain the construction of more realistic models of declarative memory [3]. To the extent that differences traditionally used to posit dual-process accounts of memory can be accounted for using simpler models like ours, the need to proliferate neuroanatomical and process-level distinctions for various memory operations can be concomitantly reduced.

The distinction between recall and recognition memory also has important implications for the presumed architecture of machine learning systems. Modern ML systems increasingly rely on a combination of distributed representation of sensory information (using deep nets) and state-centric representation of utility information (using reinforcement learning) to achieve human-like learning and transfer capabilities, for example in simple Atari games [10]. The elicitation of class or category membership in neural networks is quintessentially a recognition task, while the elicitation of state value functions, as well as other intermediate computations in RL are clearly recall tasks. Partly in realization of the large differences in the sort of memory required to support these two classes of learning models, researchers have taken to postulating dual-process artificial memories [7]. Our demonstration of the fundamental unitarity of the two modes of memory performance can and should constrain the design of deep RL models in simpler ways.

## Footnotes

[1]Our results are relatively independent of the choice of $n$, since for any value of $n$, recognition stays easier than recall so long as the cue-item fan out remains large and vice versa.

[2]Recall trials that timed out by not returning a sample beyond the maximum time limit (100 samples) are not plotted. These corresponded to 55% of the trials, resulting in a recall hit rate of 45%. In contrast, the average recognition hit rate was 82% for this simulation.

[3]An active weblink to the actual experiment is available online at [anonymized weblink].

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
