[Reviews · NeurIPS 2017]

Reviewer 1



The authors suggest a unifying mechanism to explain a variety of results on recall vs recognition memory based on the idea that in recall items are retrieved with a list as a cue and in recognition a list is retrieved with an item as a cue. It is an interesting and to my knowledge novel idea, which may contribute towards a better understanding of neuropsychological and computational processes underlying memory. However, the paper also has several weaknesses: - Most importantly, the explanations are very qualitative and whenever simulation or experiment-based evidence is given, the procedures are described very minimally or not at all, and some figures are confusing, e.g. what is "sample count" in fig. 2? It would really help adding more details to the paper and/or supplementary information in order to appreciate what exactly was done in each simulation. Whenever statistical inferences are made, there should be error bars and/or p-values. - Although in principle the argument that in case of recognition lists are recalled based on items makes sense, in the most common case of recognition, old vs new judgments, new items comprise the list of all items available in memory (minus the ones seen), and it's hard to see how such an exhaustive list could be effectively implemented and concrete predictions tested with simulations. - Model implementation should be better justified: for example, the stopping rule with n consecutive identical samples seems a bit arbitrary (at least it's hard to imagine neural/behavioral parallels for that) and sensitivity with regard to n is not discussed. - Finally it's unclear how perceptual modifications apply for the case of recall: in my understanding the items are freely recalled from memory and hence can't be perceptually modified. Also what are speeded/unspeeded conditions?

Reviewer 2



This is a beautifully written paper on a central aspect of human memory research. The authors present a simple model along with a clever experimental test and simulations of several other benchmark phenomena. My main concern is that this paper is not appropriate for NIPS. Most computer scientists at NIPS will not be familiar with, or care much about, the intricate details of recognition and recall paradigms. Moreover, the paper is necessarily compact and thus it's hard to understand even the basic descriptions of phenomena without appropriate background. I also think this paper falls outside the scope of the psych/neuro audience at NIPS, which traditionally has focused on issues at the interface with machine learning and AI. That being said, I don't think this entirely precludes publication at NIPS; I just think it belongs in a psychology journal. I think the authors make some compelling points addressing this in the response. If these points can be incorporated into the MS, then I would be satisfied. Other comments: The model doesn't include any encoding noise, which plays an important role in other Bayesian models like REM. The consequences of this departure are not clear to me, since the models are so different in many ways, but I thought it worth mentioning. In Figure 2, I think the model predictions should be plotted the same way as the experimental results (d'). I didn't understand the point about associative recognition in section 5.3. I don't think a model of associative recognition was presented. And I don't see how reconsolidation is relevant here. Minor: - Eq. 1: M is a set, but in the summation it denotes the cardinality of the set. Correct notation would be m \in \mathcal{M} underneath the summation symbol. Same issue applies to Eq. 2. - I'm confused about the difference between M' and M_k. - What was k in the simulations?

Reviewer 3



This Bayesian model is rather simple and intuitive. It gives an explanation for the recognition and recall of memory. The model and experiments can only explain short-term working memory not the long-term declarative memories. This paper introduce a simple mode for recognition and recall. It provides an Bayesian explanation for short-term working memory. However, recognition and recall normally involve long-term declarative memory, such as semantic or episodic memory, which this model can not give a reasonable explanation. The experiments can only explain the working memory from a Bayesian perspective, not the declarative memory as claimed by the authors. This paper might be more suitable for a publication at a cognitive conference /journal.